# Traversable Region Detection and Tracking for a Sparse 3D Laser Scanner for Off-Road Environments Using Range Images

**DOI:** 10.3390/s23135898

**Published:** 2023-06-25

**Authors:** Jhonghyun An

**Affiliations:** School of Computing, Gachon University, Seongnam-si 1332, Gyeonggi-do, Republic of Korea; jhonghyun@gachon.ac.kr

**Keywords:** 3D, laser scanner, LIDAR, traversability, traversable region, detection, tracking, autonomous driving, unmanned ground vehicle (UGV), off-road, range image, Bayesian fusion

## Abstract

This study proposes a method for detecting and tracking traversable regions in off-road conditions for unmanned ground vehicles (UGVs). Off-road conditions, such as rough terrain or fields, present significant challenges for UGV navigation, and detecting and tracking traversable regions is essential to ensure safe and efficient operation. Using a 3D laser scanner and range-image-based approach, a method is proposed for detecting traversable regions under off-road conditions; this is followed by a Bayesian fusion algorithm for tracking the traversable regions in consecutive frames. Our range-image-based traversable-region-detection approach enables efficient processing of point cloud data from a 3D laser scanner, allowing the identification of traversable areas that are safe for the unmanned ground vehicle to drive on. The effectiveness of the proposed method was demonstrated using real-world data collected during UGV operations on rough terrain, highlighting its potential as a solution for improving UGV navigation capabilities in challenging environments.

## 1. Introduction

Unmanned ground vehicles (UGVs) are mission-critical assets designed to operate in hazardous or harsh environments, where human intervention may be impractical, too dangerous, or infeasible, such as complex and hostile environments, characterized by rugged terrain, extreme weather conditions, or enemy threats. They are essential for various military operations, including reconnaissance, surveillance, target acquisition, and weapon delivery.

The detection of clear road boundaries and relatively flat surfaces is critical for commercial autonomous vehicles, which primarily operate on roads. Road curbs, lanes, and other structures are typically used to determine traversable areas [1,2,3,4,5,6,7]. Elevation mapping is also used to detect flat surfaces and estimate the location of vehicles relative to the ground [8,9,10,11,12,13,14,15,16,17]. Obstacle detection is another critical aspect, and previous methods have focused on identifying obstacles, to determine free spaces that are suitable for vehicle traversal [18,19,20,21,22]. These methods identify the areas where no physical obstacles directly impede the progress of the ego vehicle as traversable [23,24,25,26,27,28,29,30].

As noted in [31], the operating environment of a UGV is significantly different from that of a commercial autonomous vehicle. In such an environment, road boundaries are not always clear, and surfaces are often uneven under off-road conditions, making determining traversable regions challenging. In addition, the terrain is not paved, and road elevations can vary significantly. In contrast to the case in commercial autonomous driving, detecting structures such as road boundaries or relatively flat terrain to identify traversable regions is not always possible for a UGV. For a UGV, traversable areas refer to locations where the vehicle can move from any starting point to a target destination, without encountering obstacles or topographical restrictions. Therefore, a different approach is required to detect traversable regions for UGVs compared to that for commercial autonomous vehicles.

Traditionally, image sensors have been the primary type of sensor used to detect traversable regions. However, with the recent development of deep learning models, recognizing various scenes based on texture and other features has become possible [32,33,34,35,36,37]. These methods are a type of image analysis in which pixels corresponding to a particular object are classified into classes of objects and the target object is divided into meaningful units. In this approach, ground areas are classified based on their texture, such as mud, grass, or bushes. Traversable regions are then recognized as specific classes, based on the results of the classification process. However, these methods only identify areas that contrast with the surrounding environment in color images and ground texture. They do not provide information about whether driving in these areas is possible. In addition, this is limited in natural environments, where the illumination intensity changes rapidly due to the limitations of image sensors.

To address the limitations of current approaches to determining traversable regions, an alternative methodology employing 3D laser scanner technology has been proposed [38,39,40]. The use of 3D laser scanners enables highly precise and accurate measurements of the surrounding environment, including the ground surface, without being affected by texture or color. By analyzing the resulting point cloud data, specific criteria such as surface roughness, slope, or height differences can be utilized to identify traversable regions. This approach provides a reliable and efficient means of determining traversable regions, particularly in off-road conditions where road boundaries are not clearly defined and surfaces are uneven.

Therefore, this paper proposes a real-time traversable region-detection method using a 3D laser scanner. For real-time processing, a method for converting the 3D point cloud data into 2D images is used [12,13]. Previous studies have demonstrated the effectiveness of these approaches for accurately detecting traversable regions for paved roads in real-time processing. However, for unpaved and rough terrain, a different approach is necessary [41]. In this context, [41] processed point cloud data into a 2D range image and generated, not only vertical angle images, but also horizontal angle images, to detect the traversable regions for ego vehicles. Consequently, this method exhibited a commendable performance even in open-field environments. However, in environments where the vehicle’s pose undergoes significant changes due to the terrain, relying on a single frame for detecting traversable regions is inadequate. To overcome this challenge, a traversable region tracking method is proposed, which accumulates the detection results from previous frames. The confidence value of each pixel in the range image is leveraged to model its traversability. These confidences are then accumulated over consecutive frames using the Bayesian fusion method [42,43]. The contributions of this work can be summarized as follows:An effective traversable-region-detection method using a 3D laser scanner is proposed. To deal with a large amount of 3D point-cloud data, we used range images with each pixel indicating the range data of a specific space. Then, each pixel and the adjacent pixels are searched based on the vertical and horizontal inclination angles of the ground;A traversable-region-tracking algorithm was developed to integrate the previous detection results, to prevent detrimental effects from an unexpected pose of the vehicle. By modeling the range data of each pixel as a probability value, the traversability of the previous and current pixels in the traversable region detection results can be fused using the Bayesian fusion method.

The rest of this paper is structured as follows: In Section 2, we discuss related works in the field. Section 3 outlines the theoretical formulation of the proposed method. Moving on to Section 4, we describe the dataset configuration and the data logging system, and present the experimental results. Finally, in Section 5, we provide a summary of the paper.

## 2. Related Work

Three-dimensional laser scanner technology has been the subject of considerable attention in relation to autonomous vehicles. These scanners are widely used for environmental recognition purposes, such as object detection, map building, and route finding. In particular, the problems associated with map building can be divided into structure detection problems, such as surrounding buildings, and traversable region detection problems. When a 3D laser scanner is used for traversable region detection, many methods are available, depending on how the raw data are processed.

Thrun et al. [44] introduced a grid-based approach that divides grid cells into the ground and non-ground cells based on the height differences between points inside the cell. Moosmann et al. [8] used raw data to create a graph obtained from triangulation and utilized the concept of local convexity between two neighboring nodes in the graph. If the center point of the two connected nodes lies below the surface of the points, these points are clustered on the same ground. Himmelsbach et al. [9] represented all raw point-cloud data in a 2D x-y plane and divided this into discrete pieces of a circle. They then created each piece as a bin and identified inliers as traversable regions. Douillard et al. [45] proposed GP-INSAC, a Gaussian-process-based iterative method that classifies all raw points as the ground, with small variations in height relative to the mean of the Gaussian distribution. Chen et al. [11] presented individual raw scan data instances as a circular polar grid divided into segments. To distinguish the ground points, Chen applied a 1D Gaussian-process-based regression method for each segment, similarly to with a 1D bin. Babahajiani et al. [15] and Lai and Fox [46] used prior ground knowledge to dictate a set of all points and applied common plane fitting techniques, such as the random sample consensus (RANSAC) algorithm. Zermas et al. [16] proposed a multi-model plane fitting algorithm that divides raw data into a number of segments along the horizontal direction. The performances of these methods in terms of detecting traversable regions using all points have been verified for various specializations over many years. In addition, for improved time efficiency, computational optimization has been performed, according to the driving environment. However, searching for all the points remains inefficient with regard to time. In addition, certain model-based regression methods cannot sufficiently represent the actual surface, because the ground point does not form a perfect plane, and significant noise is generated over long distances because of raw data from 3D laser scanner measurements.

Alternatively, the raw 3D point data of the laser scanner can be projected onto a cylinder whose axis is the scanner’s axis of rotation, as opposed to using the raw data in isolation. This projection creates a range image in which the pixel value corresponds to the distance measurement. Basic research work on this aspect was conducted by Hoover et al. [47], and subsequent key approaches to local surface fitting and clustering are still being applied today. Based on this method, Bogoslavsky and Stachniss [12,13] proposed an efficient ground-search method based on range images. The ground slope is calculated using the distance between each pixel in the range image. If there is a similar slope between each pixel, the pixel is detected as the ground. However, this method assumes that the ground is a well-paved flat area in a city and it employs a means of fixing changes in the slope values. In addition, because detection is performed every moment, the information detected in the previous frame is not used.

In addition, image sensors are commonly used for detecting traversable regions. Recent advancements in deep learning models have made it possible to recognize different scenes based on texture and other features [32,33,34,35,36,37]. These methods involve image analysis techniques that classify pixels into object classes and the target objects into meaningful units. S. Palazzo et al. [35] introduced a deep-learning-based approach that estimates and predicts the traversability score of different routes captured by an onboard RGB camera. S. Hosseinpoor et al. [36] presented a method based on semantic segmentation, where they adapted Deeplabv3+ using only an RGB camera. They fine-tuned a pretrained network originally trained on Cityscapes with their own dataset. T. Leung et al. [37] proposed a hybrid framework for analyzing traversability that utilizes both RGB cameras and LiDAR. An RGB camera is used to gather semantic information, identifying different types of terrain, such as concrete or grass. Meanwhile, the LiDAR sensor provides geometric information, by creating an elevation map of the surrounding terrain.

## 3. Proposed Method

The term “traversable regions” refers to ground areas that allow a specific vehicle to move toward a target point without any obstacles. However, unmanned ground vehicles (UGVs) face more complex challenges. These include navigating through plants and grassy areas that can obstruct their movement, as well as the absence of clear structures that help distinguish between drivable and non-drivable terrain. Additionally, the presence of varying slopes makes it difficult to visually identify traversable areas. Hence, it is crucial to develop a new method for identifying traversable areas that are specifically tailored for UGVs, distinctly from the approaches used for commercial vehicles. Therefore, this paper presents a new method that uses a 3D laser scanner as the primary sensor for detecting and tracking traversable regions, as shown in Figure 1.

### 3.1. Range-Image-Based Traversable Region Detection

#### 3.1.1. Range Image

A 3D laser scanner is a device that measures the spatial information of objects or environments using laser beams, while rotating 360 degrees. It emits laser beams onto the target surfaces and measures the time it takes for the laser to bounce back to the scanner, allowing for the calculation of distances. Moreover, by maintaining a consistent vertical angle, the scanner ensures that the measurements are taken from the same reference plane throughout the entire rotation, as shown in Figure 2. These planes are called layers, i.e., 16, 32, 64, or 128 for Ouster scanners. As the scanner rotates, it emits laser beams and records the reflections from the surrounding objects at various angles. Therefore, the point cloud set $Z_{t}$ measured at time t is defined as follows:(1)$$Z_{t}=\left\{z_{t}^{1},z_{t}^{2},z_{t}^{3}\ldots z_{t}^{P}\right\}$$
(2)$$z_{t}^{k}=\left(r_{t}^{k},\theta_{t}^{k},\ell_{t}^{k}\right)$$
where the total number of measurement points is denoted by *P*; and $r_{t}^{k}$, $\theta_{t}^{k}$, and $\ell_{t}^{k}$ denote the *k*th distance, bearing, and layer at time *t* of the measurements, respectively. As such, a laser scanner provides individual layers per laser beam as raw data, along with timestamps and bearings. This facilitates the direct conversion of the raw point cloud into range images. Therefore, the range image *I* is a function.
(3)$$I:U_{I}\to \left[0,r_{max}\right]$$
(4)$$U_{I}=\left[\left[0;m-1\right]\times \left[0;n-1\right]\right]$$
are the pixels of the range image, where *m* is the number of rows in the range image defined by the number of layers in the vertical direction and *n* is the number of columns given by the range readings per 360-degree revolution of the scanner, i.e., $n=360/\Delta \theta_{h}$ respectively. $\Delta \theta_{h}$ is the horizontal resolution of the laser scanner. Therefore, a specific pixel I(i,j) has a distance value $r_{i,j}$ corresponding to the space. Therefore, the amount of data to be processed is reduced by the resolution of the range image in the form of millions of point clouds.

#### 3.1.2. Traversable Region Detection

Using the range image, we detect traversable regions by considering the vertical and horizontal inclinations of the ground. Vertical inclination refers to the slope of a path that enables the ego vehicle to move in its direction of travel, whereas horizontal inclination refers to the slope of a path that allows lateral movement based on the vehicle’s direction of travel. This enables the identification of flat ground suitable for vehicle movement. To achieve this, we first calculate the angles between consecutive rows in the range image, as follows:(5)$$\alpha_{r,c}=\mathrm{atan}2\left(\Delta x_{\alpha },\Delta z_{\alpha }\right),$$
where
(6)$$\begin{array}{c} \Delta x_{\alpha }=\left|I_{r-1,c}\sin\theta_{r-1}-I_{r,c}\sin\theta_{r}\right|,\\ \Delta z_{\alpha }=\left|I_{r-1,c}\cos\theta_{r-1}-I_{r,c}\cos\theta_{r}\right|. \end{array}$$

Additionally, to determine the horizontal inclination of the ground, the angles between consecutive columns are calculated using the range image, as follows:(7)$$\beta_{r,c}=\mathrm{atan}2\left(\Delta x_{\beta },\Delta y_{\beta }\right),$$
where
(8)$$\begin{array}{c} \Delta x_{\beta }=\left|I_{r,c}\cos\Delta \theta_{c}-I_{r,c-1}\right|,\\ \Delta y_{\beta }=\left|I_{r,c}\sin\Delta \theta_{c}\right|. \end{array}$$

Consequently, we can treat all stacks of vertical and horizontal inclination angles as range images, so we define them as angle images $M_{\alpha }$ and $M_{\beta }$:(9)$$M:U_{M}\to \left[-\frac{pi}{2},+\frac{pi}{2}\right],$$
where
(10)$$U_{M_{\alpha }}=\left[\left[0;m-2\right]\times \left[0;n-1\right]\right],$$
(11)$$U_{M_{\beta }}=\left[\left[0;m-1\right]\times \left[0;n-2\right]\right].$$

Hence,  Figure 3 illustrates the alpha angle and beta angle used for calculating the vertical and horizontal angle images. However, a mechanically rotating-type 3D laser scanner generates a significant number of outliers in the range measurements, which can affect the calculation of the angle α. The alpha angle represents the slope of the ground at the sensor’s location in the direction of the vehicle’s movement. As a result, it is heavily influenced by the vehicle’s attitude. This is discussed in more detail in the work by Leonard et al. [48]. To address this issue, Bogoslavsky et al. applied a Savitzky–Golay filter to every column of the vertical angle range images, to fit a local polynomial of a given window size to the data [13]. This filter is a smoothing method that constructs a polynomial regression model for a short signal interval within a window, as applied to a continuous signal [49]. However, column-wise smoothing using a Savitzky–Golay filter is only suitable for flat paved roads and not unpaved roads with varying heights and curves. Therefore, as shown in Figure 4, the angle image exhibits salt-and-pepper noise owing to the condition of unspecified road surfaces.

In order to remove outliers from the range measurements obtained by the 3D laser, a median filter was employed, as depicted in Figure 4. To account for a specific pixel and its surrounding pixels in the vertical and horizontal directions, a 5 × 5 kernel was used. The actual and range images are shown in the first row, while the second row shows the vertical and horizontal angle images calculated based on the angle difference between consecutive range images in the same frame, respectively. The images obtained after noise removal using a median filter are shown in the third row. Vertical angle images highlight vertical obstacles, such as vehicles, whereas horizontal angle images provide details regarding the road surface conditions, for distinguishing horizontal obstacles, such as curbs.

Finally, we propose the detection of traversable regions using the noise-removed vertical and horizontal angle images. To achieve this, the breadth-first search (BFS) method is employed from the lowest row of the range image, which is considered the ground closest to the ego vehicle, and the adjacent pixels near the four neighborhood pixels are searched. However, when calculating the angle difference between a specific pixel (i,j) and its adjacent pixel (i±1,j±1), both the vertical angle image $M_{\alpha }$ within a difference Δα and horizontal angle image $M_{\beta }$ within a difference Δβ are considered, to determine if they fall within a specific angle range. Using this process, the pixels with small differences between the vertical and horizontal angles are identified as the traversable regions. Therefore, the result of traversable region detection is $M_{T}$.

### 3.2. Probabilistic Traversable Region Tracking

In this section, we propose a probabilistic traversable region tracking method that utilizes Bayesian fusion. First, the confidence of the traversable region detected in the current frame is calculated in pixel units of the range image. Then, the confidence of each pixel is converted into class information, indicating whether the traverse is possible or not; this is also calculated in the pixel units of the range image. Finally, by applying Bayesian fusion to the accumulated class information, the traversable area can be tracked for each individual pixel within the range image, as shown in Figure 5.

#### 3.2.1. Confidence of Traversability

In the previous section, we detected the traversable region based on an angle image. However, to convert a traversable region in a single frame into a binary class of pixel units, the region must be converted into a probabilistic expression based on a specific value. The probability that a pixel belongs to a traversable region depends on the slope of the ground. The higher the slope of the ground, the lower the probability that it is a traversable region, and vice versa. Therefore, the confidence that a traverse is possible is calculated based on the difference in the height of the ground between a specific pixel and its surrounding pixels. To calculate the confidence of the traversable region for each pixel in the range image in the current frame, the following steps are performed:  
(12)$$p\left(M(i,j)\right)=1-\frac{1}{1+e^{-k(x-\varphi )}},$$
where *x* denotes the average angular difference between a particular pixel i,j in the range image and an adjacent pixel i±1,j±1. In addition, φ and *k* are hyperparameters, where φ is the midpoint for slope determination and *k* is the logistic growth rate of the range of drivable inclination. Therefore, when the difference between a specific pixel (i,j) and its adjacent pixels (i±1,j±1) in the vertical and horizontal angle image $M_{\alpha }$, $M_{\beta }$ is within a certain angle range Δα and Δβ, the ground confidence is modeled as increasing. In this study, all the pixels in the range images were defined as binary classes, i.e., either traversable or non-traversable regions.
(13)$$c_{i,j}=\left\{T,NT\right\}.$$

Therefore, the class confidence of the traversable region predicted by the current range image can be expressed as
(14)$$p\left(c=k|M\left(i,j\right)\right)=p_{k},\;\mathrm{where}\;\sum_{k\in \{T,NT\}}p_{k}=1,$$
where *c* denotes the class of traversability, *k* is one of the possible classes that *c* can take, and M(i,j) is a specific pixel in the range image. Finally, the class is predicted using
(15)$$class\left(M\left(i,j\right)\right)=\underset{k\in \left\{T,NT\right\}}{\arg\max p_{k}}.$$

#### 3.2.2. Bayesian Fusion in a Sequence

In this section, we propose a Bayesian fusion method for tracking the class information in a sequence of range images. $C_{i,j}^{1:t}$ is a sequence of the traversable region classes detected in a range image at the time up to *t*. We determine the class of the traversability sequence of the specific pixel of $C_{i,j}^{1:t}$ using maximum a posteriori (MAP) estimation. That is, the class of the sequence of the traversability is predicted using
(16)$$class\left(C_{i,j}^{1:t}\right)=\underset{k\in \left\{T,N\right\}}{\arg\max}p\left(c_{i,j}=k\left|C_{i,j}^{1:t}\right.\right).$$

Assuming that *t* denotes the current time, $C_{i,j}^{1:t}$ is divided into the current measurement $C_{i,j}^{t}$ and all previous measurements $C_{i,j}^{1:t-1}$ based on the Markov property,
(17)$$p\left(c=k|C^{1:t}\right)=p\left(c=k|C^{t},C^{1:t-1}\right).$$

This indicates that the measurements up to time t−1 have no impact on the measurements at time *t*. This is because the traversability has been accumulated for each individual pixel within the range image using the data up to time t−1. However, it is important to consider that when the traversable probability from time t−1 is propagated to time *t*, it influences the pixel-level probability values. Therefore, the probability of a traversable region in a specific pixel of a range image can be rewritten as follows using the Bayes rule:(18)$$p\left(c=k|C^{1:t}\right)=\frac{p\left(C^{t}|c=k,C^{1:t-1}\right)p\left(c=k|C^{1:t-1}\right)}{p\left(C^{t}|C^{1:t-1}\right)}.$$

This is based on the Bayes rule, $p\left(x|y\right)=p\left(y|x\right)\cdot p\left(x\right)/P\left(y\right)$. As the previous measurements $C^{1:t-1}$ do not affect the current measurement $C^{t}$ and the current measurement is conditioned on c=k, we can obtain $p\left(C^{t}|c=k,C^{1:t-1}\right)$ as $p\left(C^{t}|c=k\right)$. For the sake of simplicity, the subscripts *i* and *j*, which indicate the coordinates of a specific pixel, are omitted in the following formulas. Subsequently, by applying the Bayes rule,  
(19)$$p\left(c=k|C^{1:t}\right)=\frac{p\left(c=k|C^{t}\right)p\left(C^{t}\right)}{p\left(c=k\right)}\frac{p\left(c=k|C^{1:t-1}\right)}{p\left(C^{t}|C^{1:t-1}\right)}.$$

At this time, the sum of the class confidence values of a sequence for the binary class is 1; thus, dividing Equation (Equation 19) by the sum of the class confidence values of a sequence is mathematically equivalent to
(20)$$p\left(c=k|C^{1:t}\right)=\frac{p\left(c=k|C^{1:t}\right)}{\sum_{k^{\prime }\in \left\{T,N\right\}}p\left(c=k^{\prime }|C^{1:t}\right)}.$$Substituting Equation (Equation 19) into Equation (Equation 20) yields
(21)$$\begin{array}{c} \begin{array}{c} p\left(c=k|C^{1:t}\right)=\frac{\frac{p\left(c=k|C^{t}\right)p\left(C^{t}\right)}{p\left(c=k\right)}\frac{p\left(c=k|C^{1:t-1}\right)}{p\left(C^{t}|C^{1:t-1}\right)}}{\sum_{k^{\prime }\in \left\{T,N\right\}}\frac{p\left(c=k^{\prime }|C^{t}\right)p\left(C^{t}\right)}{p\left(c=k^{\prime }\right)}\frac{p\left(c=k^{\prime }|C^{1:t-1}\right)}{p\left(C^{t}|C^{1:t-1}\right)}}. \end{array} \end{array}$$By canceling each term, we obtain the following equation:(22)$$\begin{array}{c} \begin{array}{c} p\left(c=k|C^{1:t}\right)=\frac{p\left(c=k|C^{1:t-1}\right)p\left(c=k|C^{t}\right)p\left(c=k^{\prime }\right)}{\sum_{k^{\prime }\in \left\{T,N\right\}}\left\{p\left(c=k^{\prime }|C^{1:t-1}\right)p\left(c=k^{\prime }|C^{t}\right)p\left(c=k\right)\right\}}. \end{array} \end{array}$$

From Equation (Equation 22), we can update the sequence confidence $p\left(c=k|C^{1:t}\right)$ at time *t* from the previous sequence confidence $p\left(c=k|C^{1:t-1}\right)$ at time t−1, and the current confidence of traversability $p\left(c=k|C^{t}\right)$ directly. Thus, we do not need to retain all previous frames. Additionally, $p\left(c=k\right)$ is the initial confidence of traversability. The overall process of traversable region detection and tracking is summarized in Algorithm 1.
**Algorithm 1** Traversable Region Detection and Tracking**Input:** 3D point cloud $Z_{t}$ and previous Traverable Region $C^{1:t-1}$**Output:** Traversable Region Probability $C^{1:t}$**for every frame** *t* **do**   01: $I_{t}$← Make Range Image $\left(Z_{t}\right)$   02: $M_{\alpha }$← Make Vertical Angle Image $\left(I_{t}\right)$   03: $M_{\beta }$← Make Horizontal Angle Image $\left(I_{t}\right)$   04: $M_{T}$← Traversable Region Detection $\left(M_{\alpha },M_{\beta }\right)$   05: $C^{t}$← Traversable Confidence $\left(I_{t},M_{T}\right)$   06: $C^{1:t}$← Tracking Traversable Region $\left(C^{t},C^{1:t-1}\right)$**end for**

## 4. Experiment

### 4.1. Experiment Environment

We collected a unique dataset by conducting experiments on actual terrain. The total length of the track was approximately 1.2 km, and it contained numerous irregular slopes, thus providing a suitable environment for verifying the reliability of the proposed algorithm under various slopes and road surface conditions for a UGV. The maximum difference in the pitch angle was approximately 20 degrees, and the maximum difference in the roll angle was approximately 10 degrees, as shown in Figure 6. Based on these pitch and roll angle differences, we prepared three scenarios (routes A, B, and C) to evaluate the proposed method, using 1500 frames.

Our experimental driving platform was equipped with a 3D laser scanner, Velodyne HDL-64E sensor, and high-precision positioning system, NovAtel OEMV-2 receiver, with a Honeywell HMR3500, as shown in Figure 7. Additionally, experimental data for performance evaluation were collected by measuring a 3D point cloud using LiDAR, while simultaneously recording the vehicle’s motion information with GPS. This enabled us to collect precise route information for the driving platform in the rough terrain considered in this study.

### 4.2. Data Annotation

Defining traversable areas in places without road structures, such as road boundaries or lanes, can be challenging. In addition, even under similar road conditions, certain areas may prove difficult for vehicles to traverse. Consequently, there may be variations in the definition of traversable areas in such locations. However, annotating the ground truth based on multiple criteria may not result in good data, and evaluating the algorithm’s performance may prove difficult. To ensure consistency in annotation, this study established a unified definition of a traversable area, as an area where other vehicles have traveled or where there are visible traces of such movement that differentiate it from other areas. Data were collected by multiple annotators using this definition, although different preferences might have resulted in different traversable regions.

To collect data in point units, over 100,000 points per frame must be collected. However, the manual labeling of each point is time-consuming. Hence, in this study, a pixel-wise annotation was conducted on the range image derived from the raw 3D point cloud, as shown in Figure 8. To label the traversable regions, the annotators needed to observe the variations in the range image across consecutive frames, while referring to the corresponding actual driving images. The identified traversable areas were then represented by polygons, as shown in the figure. To ensure accuracy, data were collected by multiple personnel in the same frame, and the intersection of the areas designated by each annotator was used as the final ground-truth dataset of the traversable areas.

### 4.3. Evaluation Metrics

Labeling every point to evaluate an algorithm is time-consuming, and visualizing the traversable region in the form of a point cloud is challenging. Therefore, in this study, the ground-truth data were collected in a pixel-wise manner in the range image, as shown in Figure 8. At this point, the balance between traversable and non-traversable regions within a single-range image may vary depending on the scenario. However, these imbalanced issues in a single-range image are common in real-world situations. Figure 9 illustrates that the traversable region (represented by the gray color) occupies a smaller proportion in the range image compared to the non-traversable region (depicted in black color). To address this issue and properly evaluate both the proposed and previous methods, two evaluation metrics were used: the Jaccard index (also known as intersection-over-union (IOU)) and the Dice coefficient (also known as F1 score).

The Jaccard index is one of the most commonly used metrics in semantic segmentation and is an effective indicator. It can be calculated as the area of overlap between the predicted union of the predicted segmentation and the ground truth:(23)$$IoU=\frac{TP}{TP+FP+FN}.$$

The Dice coefficient is a statistical measure employed to assess the similarity between two samples. It is commonly used as a metric to quantify the overlap between the predicted and ground-truth segmentations.
(24)$$Dice=\frac{2TP}{2TP+FP+FN}.$$

### 4.4. Quantitative Result

To evaluate the performance of the proposed method in detecting and tracking traversable regions, we conducted a comparative evaluation with three real-time computational methods using the entire dataset. The first method detects drivable areas using an elevation-based grid map [38], in which the slopes between cells are computed using height information from a 3D laser scanner. This approach is similar to the proposed method, as it projects 3D spatial information onto a 2D grid map. The second method detects drivable areas using a range image [13], which is more effective for dealing with sparse 3D laser scanner measurements, and contains 2.5D information. This method detects traversable regions by calculating the slope of the pixels in the column direction of the range image. The third method, based on the author’s previous research, is a specialized approach designed solely for detecting traversable regions in challenging and uneven terrains [41].

The comparison results are presented in Figure 10 in the form of a range image. In the range image, each color represents an object perpendicular to the ground in different successive shades of red, with objects parallel to the ground shown in successive shades of blue. Therefore, blue indicates flat ground over which the UGV can be driven. Conversely, red indicates an impassable obstacle or non-flat ground over which the UGV cannot be driven, while gray indicates areas judged to be drivable or ground truths collected manually. The results presented in Table 1 correspond to the experiments conducted on datasets where data collection and ground truth annotation were performed. We established three types of driving paths along an approximately 1.2 km route and organized the experimental findings accordingly. Table 1 provides the conclusions in terms of the IOU and Dice values between the ground truth and predictions for each of the three path types.

The top row of the figure presents the ground truth, which was obtained by manually collecting pixel-wise images for each frame along different routes. The leftmost image corresponds to route A, and it can be observed that this route offers a relatively wide traversable area with minimal changes in the pitch angle. Most areas in this route are deemed traversable, with only a few bushes on the left and right sides serving as obstacles. The middle image corresponds to route B, which has a gentle and flat slope on the left side of the route. In the ground truth, the flat land on the left side is labeled as non-traversable, because the annotator has prior information. In fields or rough terrain, areas that are likely to be traversable often exhibit traces of previous vehicle paths, which the annotator can identify. Based on this information, only the central areas were labeled as traversable regions, while the flat land on the left side was labeled as non-traversable. However, some methods detected the flat left side of the route as a traversable region, as shown by the results. Finally, the rightmost image represents a narrow mountain road bordered by trees on both sides. Despite significant variations in pitch and roll angles, there is no vegetation present for this terrain, except for the trees. This terrain can be traversed by pedestrians, but the areas where vehicles can travel are extremely limited.

The results figure shows the outcomes for each method in the third row to the sixth row. The third row presents the results of our proposed method. The fourth row displays the results of a previous study that focused on detection without tracking. The fifth row represents the results of an elevation-map-based approach, while the sixth row shows the results of a range-image-based approach. As we can see from the experimental results, our proposed method performed the best. It achieved the most accurate detection of traversable regions for route A, which had minimal changes in vehicle pitch and roll angles and a wide traversable area. Specifically, our proposed method achieved an IoU score of 0.6701 and a Dice coefficient of 0.7971, indicating its superior performance. However, the other methods also exhibited a similar performance. In particular, for route A, which had a relatively flat terrain and minimal variations in slope, all methods demonstrated a comparable performance. The range-image-based approach, in particular, achieved a relatively high performance, with an IoU score of 0.5617 and a Dice coefficient of 0.7165. This was because the range-image-based approach assumed that the traversable areas were relatively flat. However, this method tended to mistakenly identify gentle slopes as traversable areas, which could pose a risk to the vehicle.

For route B, all methods had a lower performance overall. The other methods achieved IoU scores in the range of 0.2 and Dice coefficients around 0.3 to 0.4. However, our proposed method in this study showed an improved performance by approximately 20–30% compared to the other algorithms, with IoU and Dice coefficients of 0.2871 and 0.4269, respectively. The lower performance of all methods can be attributed to the discrepancy between the region labeled as traversable in the ground truth and the region identified as traversable based on the algorithms. This inconsistency arose because the flat land on the left side was labeled as non-traversable in the ground truth by the annotators, due to the given information that other vehicles had already passed through. Despite this discrepancy, our proposed method demonstrated a better performance than the other algorithms, because it tracked the drivable areas based on the vehicle’s trajectory. This allowed it to achieve a superior performance, as shown in the figures.

For route C, which had a narrow traversable area and many trees, the proposed method demonstrated a superior performance, with an IoU score of 0.4826 and Dice coefficient of 0.6471, compared to the other methods. Unlike routes A and B, the terrain characteristics of this route allowed us to visually identify areas that were passable. As a result, the traversable detection results of the algorithm were closer to the ground truth compared to the other routes. However, there was a potential misinterpretation of the grass existing among the trees as traversable. This was because the grass appears relatively flat compared to the trees, leading the algorithm to incorrectly detect some areas as passable. This problem was particularly pronounced with the range image-based method, which detected the entire range image as traversable.

### 4.5. Computation Time

In a subsequent experiment, we compared the real-time performance of each algorithm, considering both detection performance and operational speed as crucial factors. The run times of the algorithms were evaluated for all frames on a desktop computer with an i8-8700 3.20 GHz CPU, using only a single core of the CPU. The processing time was measured from the input of the raw point cloud data until the determination of the traversable region. As shown in Figure 11, the proposed method, the elevation-map-based method, the detection-only method, and the range-image-based method exhibited average calculation times of 2.106 ms, 19.445 ms, 2.316 ms, and 2.028 ms, respectively, for all frames. Both the range-image-based method and the detection-only method used range images, resulting in a calculation time of approximately 2 ms, since not all laser points were directly utilized. However, the proposed method required slightly more computation time compared to the range-image-based method, due to conducting searches in both the column and row directions within the range images. On the other hand, the elevation-map-based method searched through all points and required more time compared to the proposed method. Overall, these results indicated that the proposed method operated faster than the sensor’s measurement period, ensuring a real-time performance capability.

## 5. Conclusions

In conclusion, this paper presents a novel approach for detecting and tracking traversable regions using 3D laser scanners in off-road conditions. To enhance the computational efficiency, the raw data from the laser scanner, which consists of millions of data points, are processed as range images that contain distance information. Unlike previous methods that primarily focused on flat roads, our proposed approach leverages both vertical and horizontal information from range images, to robustly detect traversable regions on uneven off-road terrain. Additionally, we introduced a sequence tracking method that incorporates Bayesian fusion to integrate detection results from previous frames, ensuring resilience against abrupt changes in vehicle posture. To assess the performance of our method, we collected data while driving on an actual mountain road and obtained multiple annotations of the traversable regions in the range images. The experimental results provided compelling evidence of the effectiveness of our proposed method in real-world driving scenarios.

## Figures and Tables

**Figure 1 sensors-23-05898-f001:**
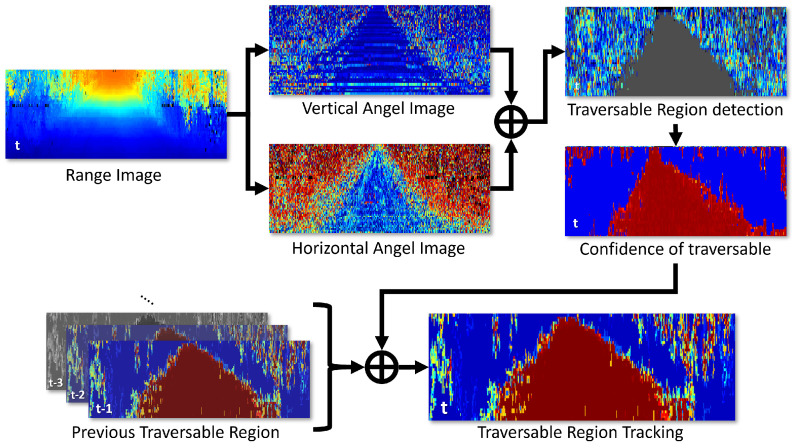
Illustration of the proposed method. In the first step, vertical and horizontal angle images are generated from the range image. Subsequently, the traversable region is detected, and the pixel-wise confidence is calculated based on this. Using this information, the traversable region is tracked using the proposed Bayesian fusion method.

**Figure 2 sensors-23-05898-f002:**
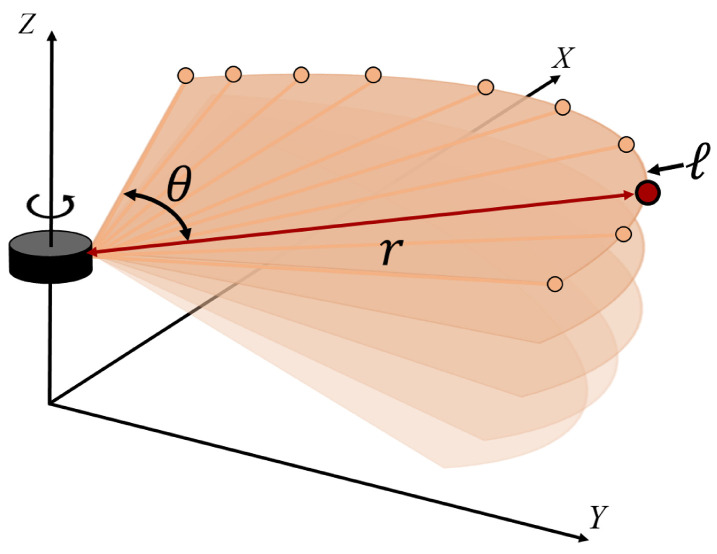
Schematic of a mechanical pulsed-time-of-flight (ToF) laser scanner.

**Figure 3 sensors-23-05898-f003:**
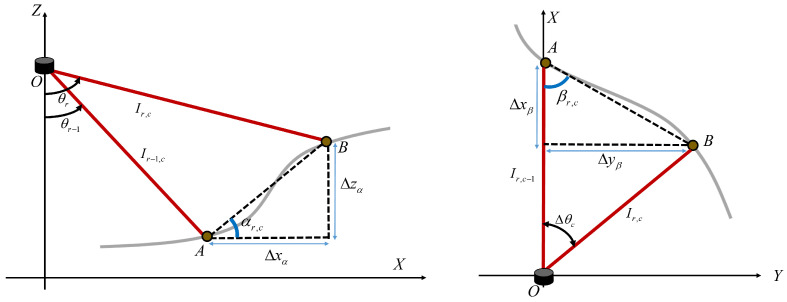
This figure illustrates the α and β angles used to compute the vertical and horizontal angle images. The red lines represent adjacent laser beams. The x-axis denotes the vehicle’s forward direction, and the z-axis represents the vertical direction perpendicular to the ground.

**Figure 4 sensors-23-05898-f004:**
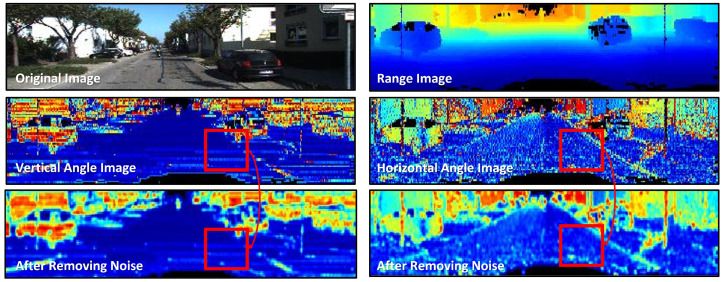
Outlier removal results in the range measurements of a 3D laser using a median filter. the second row shows the vertical and horizontal angle images, respectively. The images obtained after noise removal using a median filter are shown in the third row.

**Figure 5 sensors-23-05898-f005:**
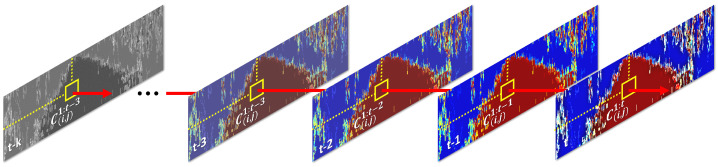
This figure presents an illustration of the pixel−wise tracking of the traversable region based on Bayesian fusion. It showcases the updated result of $C_{i,j}^{1:t}$ for the pixel (i,j) over time. $C_{i,j}^{1:t}$ denotes the sequence of the traversable region tracking results in a range image up to time *t*.

**Figure 6 sensors-23-05898-f006:**
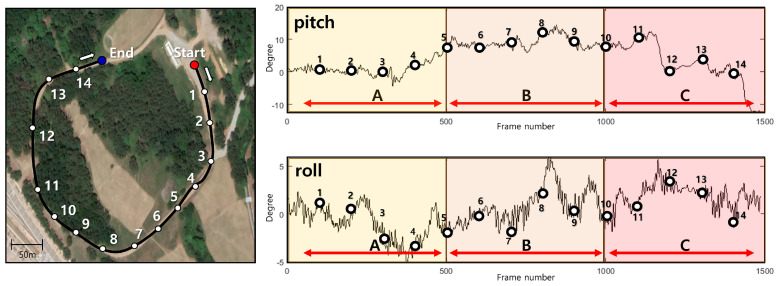
Experimental environment with actual rough terrain. (**Left**) Satellite image of the experimental route. (**Top**) Pitch angle variation with the frame number. (**Bottom**) Roll angle variation with the frame number.

**Figure 7 sensors-23-05898-f007:**
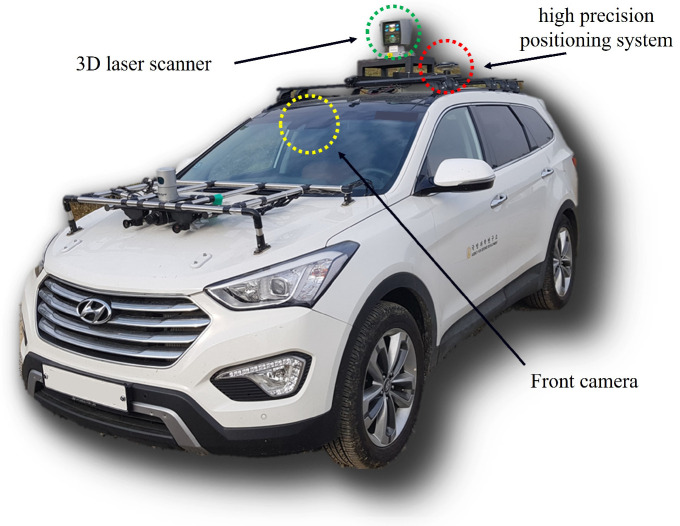
Our experimental driving platform was equipped with a high-end 3D laser scanner, a high-precision positioning system, and a front camera.

**Figure 8 sensors-23-05898-f008:**
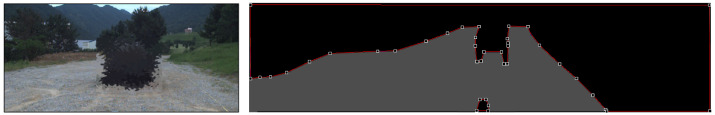
This figure shows the data annotation method. The data annotation is performed at the pixel level, marking traversable and non-traversable regions. The gray area represents the traversable region, while the black area indicates the non-traversable region.

**Figure 9 sensors-23-05898-f009:**
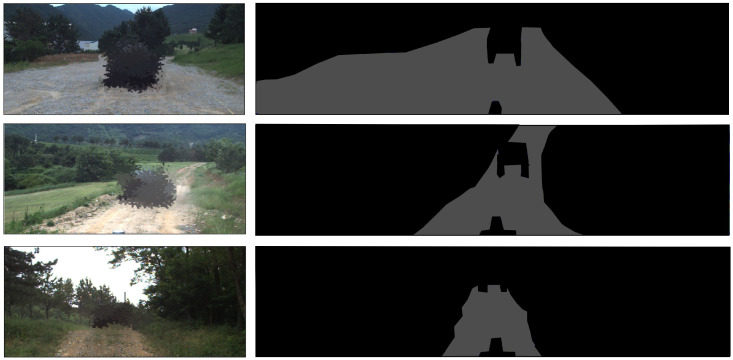
The range image, converted from a raw 3D point cloud, was annotated pixel-wise to distinguish between traversable and non-traversable regions. The images on the left show the actual environment of the rough terrain, while those on the right represent the annotation results for the traversable regions.

**Figure 10 sensors-23-05898-f010:**
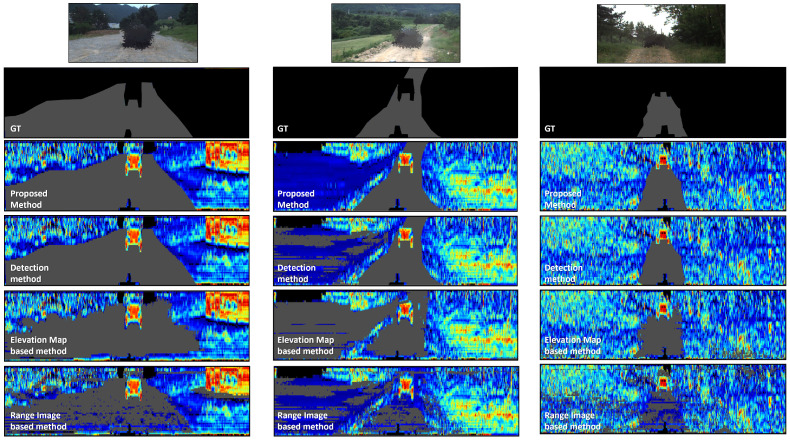
This figure showcases the experimental results. The first row exhibits the actual images, while the second row presents the labeled ground truth (GT). The experimental results are displayed as range images obtained from the raw 3D point cloud. The third row illustrates the approach proposed in this paper. Furthermore, from left to right in each column, they respectively represent the route segments A, B, and C. A comparative analysis with the other results demonstrated that the method proposed in this paper delivered improved outcomes.

**Figure 11 sensors-23-05898-f011:**
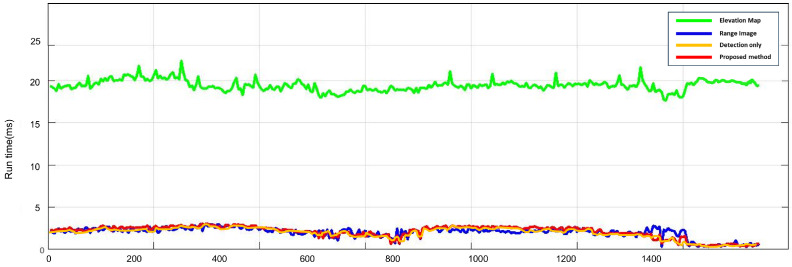
This figure represents the comparison results for computation time. The method proposed in this paper is indicated by the red line. It required a similar computation time as the other methods that utilized range images, but demonstrated a significantly higher efficiency compared to the methods using elevation maps.

**Table 1 sensors-23-05898-t001:** Quantitative Results of Traversable Region Detection.

Method	Route A		Route B		Route C	
	Iou	Dice	Iou	Dice	Iou	Dice
Elevation Map [38]	0.5090	0.6726	0.2004	0.3291	0.1610	0.2765
Range Image [13]	0.5617	0.7165	0.2069	0.3399	0.2997	0.4563
Detection only [41]	0.6509	0.7870	0.2773	0.4259	0.4816	0.6461
Proposed method	0.6701	0.7971	0.2871	0.4269	0.4826	0.6471

## Data Availability

Restrictions apply to the availability of these data. Data was obtained from Agency for Defense Development.

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
