# Peer review of "Traversable Region Detection and Tracking for a Sparse 3D Laser Scanner for Off-Road Environments Using Range Images"

_sensors, 2023, doi:10.3390/s23135898_

Round 1

Reviewer 1 Report

The author presents a traversability analysis method for offroad navigation based on 3D LIDAR scans. To improve the quality of the estimation the author proposed a Bayesian method for tracking traversable areas over time.

The paper deals with the hot topic of UGV navigation in unstructured environments, where traversability estimation is crucial. The solution proposed by the author looks promising. However, several aspects need to be clarified and addressed.

The author could mention the following works among those deep learning based solutions for offroad navigation mentioned at line 42:

https://ieeexplore.ieee.org/document/9341044

https://ieeexplore.ieee.org/document/9551629

https://ieeexplore.ieee.org/document/9738557

Lines 59-60: "In order to facilitate real-time processing, a method for converting the 3D point cloud data into a 2D image has been proposed". It seems the author refer to a method already proposed in the literature. This becomes clearer only at lines 137-139. However, for the sake of clarity it would be better to include a reference at lines 59-60.

Line 97: why does the author mention curriculum learning?

Line 126: how is the mention "x-axis" oriented?

Line 165: the author should be more specific when mentioning "the distance between a specific location"

Line 166: What does the author mean by "rings at the same elevation"? Shouldn't they be at different altitudes actually?

Lines 176-180: The author explicilty states that "m and n correspondingly denote the number of layers in the vertical direction and the range readings per 360deg revolution of the scanner". If my understanding is correct this would mean that the total number of pixels of the range image is the same of the total number P of LIDAR points. Hence, how can the author state "the amount of  data to be processed is reduced by the resolution of the range image in the form of millions of point clouds". It is actually also unclear why the author says 'millions of point clouds'. Please clarify this aspect as it is crucial.

To improve readability, I would suggest to include a graphical scheme showing where equations 5-8 come from, including the points and the angles. Still  for clarity, in Eq. 8 the author should specify what is meant by \Delta\theta.

Line 194: the author refers to Malpha and Mbeta as range images (please fix algorithm 1 where these two images are indicated as Ma and Mb). This is really misleading and makes confusion with the actual range image I. I would suggest to refer to these images as the author refers to them in other parts of the paper, i.e. as "angle images". Such a confusion is exhacerbated in lines 245 and 257.

Line 199: It is unclear why outliers affect only the calculation of the Alpha angle. Please clarify this aspect.

Line 207: it is unclear on which array of data the median filter is applied.

Line 217: the author should be more precise in defining what is the ground "close to" the vehicle, as this is the starting step of the remainder of the algorithm.

Lines 220-221: "both the vertical inclination difference \Delta\alpha and the horizontal inclination difference \Delta\beta are considered to determine whether it falls within a specific angle range." Does this mean that at the end of this step you obtain a single image including the traversable (and nontraversable) regions from the two angle images?

Lines 233-236: what does the author mean by "in pixel units"? why it is being specified just at this point?

Line 241: the author generically says "the slope of the ground". However, according to the explanation given in line 250 it seems that just the vertical angular difference is considered. Why is the information on the horizontal angle neglected?

Lines 260-261: "a sequence of the traversable region classes detected in a range image at time t". It is unclear what the author means by "sequence of traversable region classes": does the author refer to the set of pixels classified as traversable? What is the range image? The merged Malpha and Mbeta, either one of them or the actual range image I ? Moreover, for the sake of clarity the end of the sentence should be "up to time t" as all previous images must be considered.

In algorithm 1 what is Pt? Why line 4 assigns the calculaitn of traversable confidence to Ma? Finally, probably including a graphical scheme showing how the range image is processed could be clearer than such a high-level algorithm.

In Eq. 17 the author seems to make some assumptions in applying the Bayes Rule. Please provide further details on how such an equation is derived. The same comment holds for Eq. 20.

Line 275: The author states that with Eq 21 there is no "need to retain all previous frames". However it seems the term C1:t-1 does include all previous frames up to t-1. Please clarify this.

Fig. 4: the author says the proposed method performs a binary classification. Thus, I would have expected to see only a binary (i.e. black and white) image. In lines 340-346 the authors explains the meaning of the color shades, but still this is not how a ground truth for a binary classificator should look. Finally still in lines 345-346 I don't understand how the ground truth can be available at deploying time.

Line 291-292: the author says "while reducing the influence of the vehicle’s movement using the inertial measurements". Isn't such an influence addressed by the Bayesian filtering? Why the need to introduce such a sensor? How is the vehicle's movement mitigated with inertial measurments? Please discuss these points.

Lines 308-309: the author states "as depicted in Fig. 4" but no information on how data labeling is performed and furthermore it is unclear how the proposed labeling is not time-consuming as mentioned in line 307.

Line 320: it is unclear why the authors mentions the two metrics  (IoU and F1 score) to address the issue of the imbalanced dataset. These are just metrics to ensure the quality of the classification but they do not directly act on the imbalance of the examples included  in the dataset. Please correct this.

It is unclear whether the metrics results reported in Table 1 and explained from line 365 on are average values over all the range images acquired for each segment or just refer to the images shoen in Fig. 5

Lines 377-380: it is unclear why the author refers that the problem lies in the discrepancy in the data labeling even though the rationale behind such a labeling was provided and motivated in sec. 4.2.

Line 414: shouldn't the computation time evaluation include also the time required to convert the LIDAR into a range image? Is it already included?

Minors:

In eq. 13 the author introduces the two classes as T and N, but sometimes they are indicated as T and NT (lines 14-15)

Line 262: "the maximum a posteriori (MAP)" please add "estimate".

In Eq. 21 the term p(c = k') should be moved closer to the summation operator as it includes the summation index k'

In Fig. 3 two 3D LIDARs are indicated but only one is mentioned. Please clearly state the other is unused. Moreover, a front camera is shown which is not actually used for the proposed approach. Please explain the role of the front camera if any.

Line 289: please clearly state the type of sensors along with the commercial product used.

Table 1 is not mentioned in the body text

Lines 326-330: the IoU definition is repeated twice and F1 score is undefined (please check equations as well).

Fig 5: I would suggest to include the image from the real world. Please revise the caption and remove "first method" and "last method" but refer to route segments A, B, and C.

Line 414: Correct mm to ms.

The paper can definitely be improved. The wording of some sentences is confusing and potentially misleading.

Some examples:

"Measure the surrounding environment" (Line 58) or "measure space" (used in several points) in place of "range" or "distance measurement" is improper.

The author often refers to the vehicle as "host" vehicle. Why host?

Line 83: The proposed method suggests that the proposed approach is an effective means of detecting traversable regions (needs to be rephrased)

Lines 98-99: "Section 4 introduces the dataset configuration along with the data logging system and shows the experiment results in we conducted." Was logging meant to be labeling? Moreover, please recheck final part of the sentence.

Line 116: "discrete pieces of a circle"? Is it a circular sector?

Author Response

The submitted manuscript is the revised version of sensors-2428099, according to the decision

"Major Revisions".

  • Manuscript ID: sensors-2428099
  • Type of manuscript: Article
  • Title: Traversable Region Detection and Tracking for a Sparse 3D Laser Scanner for Off-Road Environments Using Range Images
  • Authors: Jhonghyun An *
  • Received: 16 May 2023
  • E-mails: [email protected]
  • Submitted to section: Sensing and Imaging,

Dear Reviewers,

Thank you very much for your time and efforts in coordinating the review process of our submitted paper. According to your suggestions and comments made by reviewers, we have addressed the raised questions and carefully revised our original manuscript.

For your convenience, changes are highlighted by the red color in the revised manuscript. The detailed reply to each comment and some additional changes are presented below. Please refer to the following replies for more details. I believe we have addressed all the concerns, and the quality of the revised version is much improved.

Many thanks again for all your kind help and hard work.

Best regards,

Jhonghyun

Reviewer 2 Report

This study proposes a method for detecting and tracking traversable regions in off-road conditions for unmanned ground vehicles. Its main contribution consists in using a 3D laser scanner and a range-image-based traversable-region-detection approach, followed by a Bayesian fusion algorithm.

The document is sometimes hard to read and follow.

The document is well supported with references although old.

The subject of the paper has great potential of application.

In line 4 please correct to “…to ensure safe…”

In line 99 please correct “…shows the experiment results in we conducted…”

Related work is very summarized. Author should present a deeper analysis including works with similar approaches.

Figure 1 is presented before it is referenced in the text. Please correct.

It is not clear the source of Figure 1. Please correct. Also, it is not clear the reason to include an image at the end of the related work which is not a summary of the related work.

In the first paragraph of Proposed Method section author start referring the UGVs as military without any explanation. Please correct.

In subsection 3.1.1. author should include an image illustrating the laser scanner used and a schematic of the laser operation mode.

At the end of subsection 3.1.2 author should include a schematic explaining the procedure of noise removal.

In line 251 authors says that “…when the difference in the vertical angle of the ground between adjacent pixels is within a certain range…”. What is that certain range? How is it determined? Please clarify in the text.

Figure 2, 3 and 4 are presented before referenced in the text. Please correct.

Authors should present a deeper explanation of the ground truth collecting procedure.

In subsection 4.4 author should present a summary table with comparison of the results of the different methods.

Please correct the yy axis labels in Figure 7.

In line 414 please correct “…approximately 2 mm…”

The English needs some spell checking.

Author Response

(The authors gave the same response as above.)

Reviewer 3 Report

This paper proposes a method for detecting and tracking traversable regions in off-road conditions for unmanned ground vehicles (UGVs). This is a well-written paper containing interesting results which merits publication. For the benefit of the reader, however, a number of points need clarifying and certain statements require further justification. There are given below.

1. It is noted that your manuscript needs careful editing by someone with expertise in technical English editing paying particular attention to English grammar, spelling, and sentence structure so that the goals and results of the study are clear to the reader.The authors must have their work reviewed by a proper translation/reviewing service before submission; only then can a proper review be performed. Some sentences contain grammatical and/or spelling mistakes or are not complete sentences.The quality of English needs improving.

2. In the introduction part of the article, sufficient references are cited, which is done very well. However, some documents are not new enough to let readers understand the current difficulties and cutting-edge progress in this field. I think it is necessary to elaborate on the difficulties of this issue in more detail and cite more cutting-edge excellent papers.

3. Below all the pictures in the article are detailed explanations about the content of the pictures, which allows readers to understand the meaning of the pictures at a glance. I think it would be better if this kind of explanation can be more streamlined, or some explanatory text can be put into the text .

4. For figure 4, can you elaborate on the criteria for distinguishing different scenarios? For example, the difference between the normal and hard scenes in the picture is not obvious enough in my opinion. If possible, I hope to have experimental data of more difficult or obvious scenes.

5. In the experimental chapter, there are detailed and sufficient experimental data and image-specific picture descriptions, but there is a lack of comparison with the most cutting-edge methods for this problem. Such problems can be solved by using the methods in this paper, and it would be better if the advantages and disadvantages of the methods described in this paper and other methods can be shown.

6. Did the author consider situations such as manhole covers on the road? If the manhole cover is lost and only an open manhole is on the road, can the camera accurately distinguish such a situation? This is just a question and does not require an answer in the article.

It is noted that your manuscript needs careful editing by someone with expertise in technical English editing paying particular attention to English grammar, spelling, and sentence structure so that the goals and results of the study are clear to the reader.The authors must have their work reviewed by a proper translation/reviewing service before submission; only then can a proper review be performed. Some sentences contain grammatical and/or spelling mistakes or are not complete sentences.The quality of English needs improving.

Author Response

Response to Reviewers

The submitted manuscript is the revised version of sensors-2428099, according to the decision

"Major Revisions".

  • Manuscript ID: sensors-2428099
  • Type of manuscript: Article
  • Title: Traversable Region Detection and Tracking for a Sparse 3D Laser Scanner for Off-Road Environments Using Range Images
  • Authors: Jhonghyun An *
  • Received: 16 May 2023
  • E-mails: [email protected]
  • Submitted to section: Sensing and Imaging,

Dear Reviewers,

Thank you very much for your time and efforts in coordinating the review process of our submitted paper. According to your suggestions and comments made by reviewers, we have addressed the raised questions and carefully revised our original manuscript.

For your convenience, changes are highlighted by the red color in the revised manuscript. The detailed reply to each comment and some additional changes are presented below. Please refer to the following replies for more details. I believe we have addressed all the concerns, and the quality of the revised version is much improved.

Many thanks again for all your kind help and hard work.

Best regards,

Jhonghyun

Round 2

Reviewer 1 Report

The reviewer sincerely appreciates tha careful revision carried out by the author. All the main concerns have been addressed.

I have just some final comments/clarifications on my questions and the related author's answer.

Point 10: it seems now that you used a convolutional median filter on the input image (hence the array of data to which such a filter is applied is nothing but the serialized array of the pixel values from the input image within the kernel). Hence, please specify the kernel size.

Point 13: it is now clearer what the author meant by "in pixel units", but I would suggest to keep the explicit wording adopted by the author themself in the reply i.e. "the confidence values are updated for each individual pixel" as in "pixel units" still sounds misleading. Probably "pixel-wise" could be a better or alternative wording.

Point 17: I would like to thank the author for including the explanation. However, I did not want the author to explain the well-known Bayesian inference steps, but to include further details on the mathematical derivation of the formulas. Probably notation could be simplified. For instance also in Point 18 the author explicitly mentions the Markov assumption. However I would have expected to read Eq. 17 as:

p(c = k|C1:t)= p(c = k|Ct, Ct−1)

Hence, I would suggest the author to further clarify their notation.

Minors: please make more readable the writings in the newly added/modified Figures 3, 5, 10 and 11.

Author Response

Response to the Reviewer

The submitted manuscript is the revised version of sensors-2428099.

  • Manuscript ID: sensors-2428099
  • Type of manuscript: Article
  • Title: Traversable Region Detection and Tracking for a Sparse 3D Laser Scanner for Off-Road Environments Using Range Images
  • Authors: Jhonghyun An *
  • Received: 16 May 2023
  • E-mails: [email protected]
  • Submitted to section: Sensing and Imaging,

Dear Reviewer,

Thank you very much for your time and efforts in coordinating the review process of our submitted paper. According to your suggestions and comments made by reviewers, I have addressed the raised questions and carefully second revised our original manuscript.

For your convenience, changes are highlighted by the red color in the revised manuscript. The detailed reply to each comment and some additional changes are presented below. Please refer to the following replies for more details. I believe we have addressed all the concerns, and the quality of the revised version is much improved.

Many thanks again for all your kind help and hard work.

Best regards,

Jhonghyun

School of Computing, Gachon University,

1342 Seongnamdaero, Sujeong-gu, Seongnam-si, Gyeongi-do, Republic of Korea.

Reviewer 2 Report

Since the author addressed the main issues pointed out in the previous review i advise that the manuscript be accepted for publication.

English is ok.

Author Response

Response to the Reviewer

The submitted manuscript is the revised version of sensors-2428099

  • Manuscript ID: sensors-2428099
  • Type of manuscript: Article
  • Title: Traversable Region Detection and Tracking for a Sparse 3D Laser Scanner for Off-Road Environments Using Range Images
  • Authors: Jhonghyun An *
  • Received: 16 May 2023
  • E-mails: [email protected]
  • Submitted to section: Sensing and Imaging,

Dear Reviewer,

Thank you very much for your time and efforts in coordinating the review process of our submitted paper. According to your suggestions and comments made by reviewers, I have addressed the raised questions and carefully second revised our original manuscript.

For your convenience, changes are highlighted by the red color in the revised manuscript. The detailed reply to each comment and some additional changes are presented below. Please refer to the following replies for more details. I believe we have addressed all the concerns, and the quality of the revised version is much improved.

Many thanks again for all your kind help and hard work.

Best regards,

Jhonghyun

School of Computing, Gachon University,

1342 Seongnamdaero, Sujeong-gu, Seongnam-si, Gyeongi-do, Republic of Korea.
